# Unstructured clinical notes within the 24 hours since admission predict short, mid & long-term mortality in adult ICU patients

**Maria Mahbub**[1,4]*, **Sudarshan Srinivasan**[1], **Ioana Danciu**[2], **Alina Peluso**[2], **Edmon Begoli**[1], **Suzanne Tamang**[3], **Gregory D. Peterson**[4]

**1** Cyber Resilience and Intelligence Division, Oak Ridge National Laboratory, Oak Ridge, TN, United States of America, **2** Computational Sciences and Engineering Division, Oak Ridge National Laboratory, Oak Ridge, TN, United States of America, **3** Center for Population Health Science, Stanford University, Palo Alto, CA, United States of America, **4** Department of Electrical Engineering and Computer Science, University of Tennessee, Knoxville, TN, United States of America

* mmahbub@vols.utk.edu

**Data Availability Statement:** Data are publicly available in https://mimic.mit.edu/. Researchers can access the data by completing the training course "Data or Specimens Only Research"

## Abstract

Mortality prediction for intensive care unit (ICU) patients is crucial for improving outcomes and efficient utilization of resources. Accessibility of electronic health records (EHR) has enabled data-driven predictive modeling using machine learning. However, very few studies rely solely on unstructured clinical notes from the EHR for mortality prediction. In this work, we propose a framework to predict short, mid, and long-term mortality in adult ICU patients using unstructured clinical notes from the MIMIC III database, natural language processing (NLP), and machine learning (ML) models. Depending on the statistical description of the patients' length of stay, we define the short-term as 48-hour and 4-day period, the mid-term as 7-day and 10-day period, and the long-term as 15-day and 30-day period after admission. We found that by only using clinical notes within the 24 hours of admission, our framework can achieve a high area under the receiver operating characteristics (AU-ROC) score for short, mid and long-term mortality prediction tasks. The test AU-ROC scores are 0.87, 0.83, 0.83, 0.82, 0.82, and 0.82 for 48-hour, 4-day, 7-day, 10-day, 15-day, and 30-day period mortality prediction, respectively. We also provide a comparative study among three types of feature extraction techniques from NLP: frequency-based technique, fixed embedding-based technique, and dynamic embedding-based technique. Lastly, we provide an interpretation of the NLP-based predictive models using feature-importance scores.

## Introduction

Intensive Care Units (ICUs) have the highest mortality rate and highest costs among all hospital units [1]. Each year in the United States, more than 5 million patients get admitted to ICUs, with a mortality rate of 10% to 29% (source: https://www.sccm.org/Communications/Critical-Care-Statistics). The critical conditions and the urgent need to stabilize ICU patients require the efficient allocation of costly ICU resources. Prediction of mortality in ICU patients can not

provided by CITI (https://about.citiprogram.org/en/homepage/).

**Funding:** This research used resources of the Knowledge Discovery Infrastructure at the Oak Ridge National Laboratory, which is supported by the Office of Science of the U.S. Department of Energy under Contract No. DE-AC05-00OR22725 and the Department of Veterans Affairs Office of Information Technology Inter-Agency Agreement with the Department of Energy under IAA No. VA118-16-M-1062. This manuscript has been in part co-authored by UT-Battelle, LLC under Contract No. DE-AC05-00OR22725 with the U.S. Department of Energy. The United States Government retains and the publisher, by accepting the article for publication, acknowledges that the United States Government retains a non-exclusive, paid-up, irrevocable, world-wide license to publish or reproduce the published form of this manuscript, or allow others to do so, for United States Government purposes. The Department of Energy will provide public access to these results of federally sponsored research in accordance with the DOE Public Access Plan (http://energy.gov/downloads/doe-public-access-plan). The funders had no role in study design, data collection and analysis, decision to publish, or preparation of the manuscript. The specific roles of all authors are articulated in the 'author contributions' section.

**Competing interests:** The authors have declared that no competing interests exist.

only assist health professionals in the clinical decision-making process but also serve as a ground for managing hospital resource utilization.

For many years, to predict ICU mortality, researchers have used point-based scoring systems such as Simplified Acute Physiology Score (SAPS) and Acute Physiology and Chronic Health Evaluation (APACHE). Despite being widely used, these techniques suffer from several implementation challenges such as definition ambiguities, classification or selection bias [2] and missing values [3]. The accessibility of electronic health records (EHR) data has the potential to revolutionize mortality prediction using data-driven machine learning approaches. [4].

Mortality prediction in the ICU has primarily focused on forecasting mortality in patients using observational data that is collected in the EHR. Two types of EHR data used for predictive modeling are: i) structured (i.e., coded data) with a predefined and consistent format such as age, demographics, ICD codes, labs, etc., and ii) unstructured data with rather irregular and unorganized form such as free-text clinical notes that are documented by a clinician during the course of patient care. While the versatility of machine learning models allows using either form of EHR, using only structured data has historically been the choice among researchers [5]. Nonetheless, similar to point-based scoring systems, modeling with structured data is also not immune to issues such as data missingness, noisiness, and irregularities, data handling bias, errors in recording equipment, and so on [6].

Unstructured data occupy approximately 80% of the EHR and have an unparalleled abundance of information [7]. In this paper, we show that the mortality of ICU patients can be predicted using only one type of unstructured EHR data: the textual clinical notes taken by the care providers. Clinical notes contain the trajectory of health conditions in patients and different treatment measures with more granularity than structured data. The examples of these clinical notes include, but are not limited to, patients' history, consultation notes, physiological conditions, laboratory report narratives, progress notes. Although there is some overlap, clinical notes may also contain information not documented in structured data, resulting in better performance of machine learning models in clinical decision-making tasks [8, 9]. Moreover, while machine learning models trained on structured data can lead to good performance, the strict formatting of the structured data may somewhat restrict the models' perception of the patients' conditions and the thought process of the care providers [10]. On the other hand, models trained on free-form clinical notes have the flexibility to capture the thought process of the healthcare providers to a greater extent and thus may better assist in making well-informed decisions [9, 11, 12]. Nevertheless, to conclude the advantages of using unstructured clinical notes over structured data for predictive modeling in our problem setting, we need to further experiment with only structured data as well as both structured and unstructured data. As the focus of this study is to unveil the potential of raw unstructured clinical notes for predicting mortality in adult ICU patients, for this study, we use solely unstructured clinical notes in prediction modeling and leave the comparative analysis for future work. While processing of these unstructured clinical notes is a necessity to identify relevant information for predictive modeling, rigorous processing may cause the features extracted from the notes susceptible to confirmation bias [13]. Hence, we use raw clinical notes for this task.

In this work, we define the outcome of the mortality prediction task by the probability of being alive. Thus, using MIMIC3 data, we propose a framework to predict short-term, mid-term and long-term mortality in adult (equal or greater than 18 years old) ICU patients using clinical notes documented within the first 24 hours of admission [14]. We use the statistical description of the patients' length of stay to define the short-term as 48-hour and 4-day period, the mid-term as 7-day and 10-day period, and the long-term as 15-day and 30-day period after admission.

As predictive models, we use four machine learning approaches: logistic regression [15], light gradient boosting machine [16], random forest [17], and a 3-layer feed-forward neural network [18]. We compare performance of each models using the area under the receiver operating characteristic (AUROC or ROC) curve, sensitivity and specificity scores. Additionally, we compare three kinds of feature extraction techniques from NLP: frequency-based TF-IDF [19], fixed embedding-based FastText [20], and dynamic embedding-based transformer encoder model [21]. We argue that each of the techniques brings different perspectives to the task. To convey information on the interpretability of our results, we also provide feature importance to justify the predictions performed by our models.

Our contribution is 3-fold as follows. First, we propose a general framework for predicting short/mid/long-term mortality in adult ICU patients that can take free-text clinical notes as input and produce the probability of patients' being deceased/alive without manual feature engineering. The end-to-end pipeline makes our framework easily adaptable to healthcare systems. Second, we identify the potential of clinical notes taken within only 24 hours after admission for the mortality prediction task. Third, we discuss the potential benefit of different feature representations and provide an interpretation of the prediction results to increase the transparency and accountability of the machine learning models. Our study focuses on addressing the potential of free-text clinical notes from *24-hours* after admission for mortality prediction in short/mid/long-term period. Thus we focus on approaching the problem with simple interpretable machine learning algorithms.

We organize the rest of the paper as follows: section *Related works* provides a brief discussion of the previous related works, section *Materials and methods* details the data and the methodology of the study, section *Results & discussion* discusses the results and finally, section *Conclusion & future works* concludes the paper with some possible directions for future research works.

## Related works

In this paper, we focus on short-term, mid-term, and long-term mortality prediction in ICU patients using unstructured clinical notes from MIMIC III [14] database. Thus, our work is in the nexus of three research areas: short/mid/long-term mortality prediction using MIMIC III data, use of machine learning models, and contribution of unstructured clinical notes in the prediction task.

Over the years, researchers have focused on different granularity of time range for in-hospital mortality prediction such as 6-hour mortality in ICU patients [22], 8 to 24-hour, 1-day and 2-day mortality in NICU patients [23], 24-hour mortality in acute myocardial infarction patients [24], 7-day mortality in acute heart failure patients [25], 28-day mortality in sepsis-induced coagulopathy patients [26], 90-Day survival in acutely ill patients [27], and so on.

A variety of linear and non-linear machine learning models have been used for mortality prediction. In [28], authors have used XGboost [29] to predict 30-day mortality in patients with sepsis and reported 0.857 AUC. The authors in [30] have also used Gradient boosting for real-time mortality prediction and achieved 0.92 AUC. Logistic regression is reported to perform well in [31–34]. Random forests, k-nearest neighbors, support vector machines, decision trees and ensemble learning are also used in [33, 35–38]. While traditional machine learning approaches have been the norm in the clinical domain for years, newer mortality prediction studies have adopted deep learning-based approaches [39–44]. In [45], authors have used self normalizing neural network to predict 30-day mortality in ICU patients with an AUC of 0.8445, whereas [46] uses long short-term memory recurrent neural network to predict 12-hour mortality in ICU patients.

While these studies mostly focus on structured data, the adaptation of unstructured clinical notes is also becoming more common [13, 47, 48]. Authors in [13] have used fastText [20] to process clinical notes and then trained a hierarchical Convolutional Neural Network-Recurrent Neural Network (CNN-RNN) for 12-hour, 24-hour, and 48-hour mortality prediction in ICU patients. In [49], the authors have used textual nursing notes and processed them using Term Frequency–Inverse Document Frequency (TF-IDF) and further trained a logistic regression model for 1-month mortality prediction in chronic kidney disease patients. They have achieved 0.782 AUC. In [50], the authors have also used unstructured clinical notes for mortality prediction. They have used word2vec [51] and bag-of-words as feature extraction methods, and XGBoost and Logistic Regression as machine learning models. Researchers have adapted the state-of-the-art *Bidirectional Encoder Representations from Transformers (BERT)* [52] model on several tasks in clinical domain such as clinical named entity recognition, medical natural language inference, de-identification, document classification, sentence similarity measurement, relationship extraction and so on [53–55].

However, the main impediment of using transformer-based models in clinical classification tasks such as mortality prediction is the truncation of clinical notes due to limited dimension of input sequence length [56]. Authors in [56] have tackled this issue by using a hierarchical CNN transformer model and achieved an AUROC of 0.78 on mortality prediction within the ICU stays.

## Materials and methods

In this section, we discuss the data, patient cohort, data processing, the proposed framework for the mortality prediction tasks, three types of feature extraction techniques, metrics to measure models' performance, and the experimental setup. As shown in Fig 1, the primary components of the framework are feature extraction and predictive modeling of the short-term, mid-term and long-term mortality.

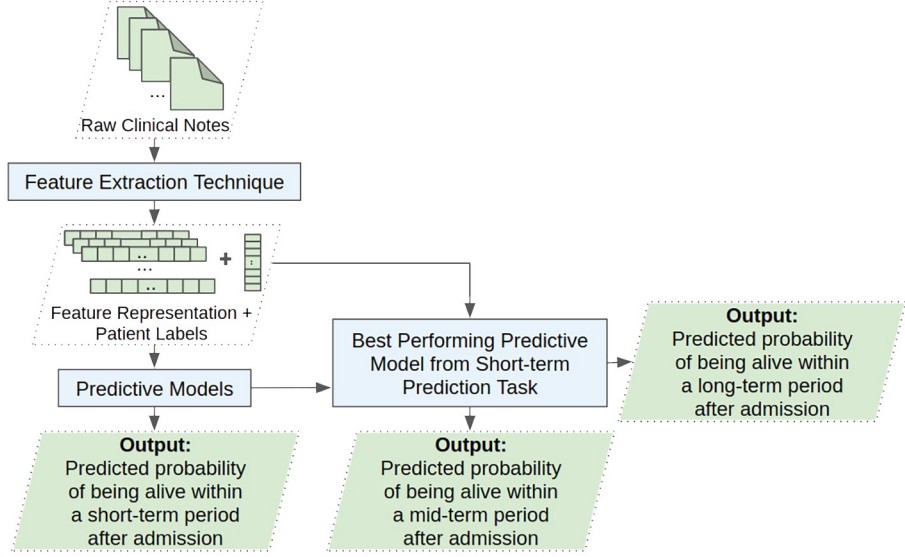

**Fig 1. Proposed framework for short-term, mid-term and long-term mortality prediction tasks.**

## Data

We use the publicly available medical database consisting of de-identified health-related data from ICU patients in the Beth Israel Deaconess Medical Center in Boston, Massachusetts between 2001 and 2012, namely MIMIC III v1.4 [14]. The database has pre-existing Institutional Review Board (IRB) approval. Hence, no further approval or consent was required. Researchers can access the data by completing the training course "Data or Specimens Only Research" provided by CITI (available at: https://about.citiprogram.org/).

**Cohort and clinical notes selection.** The MIMIC III database consists of 26 tables linked by identifiers for unique patients. For this study, we only use three tables from the database: PATIENTS, ADMISSIONS, and NOTEEVENTS. Among the 46,520 patients in the ADMISSIONS table, 46,146 patients have at least one clinical note. For patients with multiple admissions, we only keep the last one since it has the ultimate discharge or death information. As neonatal patients are not the focus of this study, we remove pediatric patients, those less than 18 years at the time of ICU admission. We calculate the age of patients using the DOB field in the PATIENTS table and the hospital admission time, ADMITTIME field in the ADMISSIONS table.

As our inputs, we use only clinical notes from the TEXT field in the NOTEEVENTS table. The clinical notes were documented during the course of patient care. For our study, we use the clinical notes from 14 out of 15 categories in the NOTEEVENTS table: Case Management, Consult, ECG, Echo (echoencephalogram), General, Nursing, Nursing/other, Nutrition, Pharmacy, Physician, Radiology, Rehab Services, Respiratory, and Social Work. We exclude the discharge summaries since discharge summaries explicitly mention death/discharge information and can cause possible data leakage.

We also remove incomplete or erroneous clinical notes. The incomplete or erroneous clinical notes are defined by the Boolean value 1 in the ISERROR field in the NOTEEVENTS table. We also remove notes that were taken before admission to maintain consistency. Fig 2 shows the detailed cohort selection process in a consort diagram.

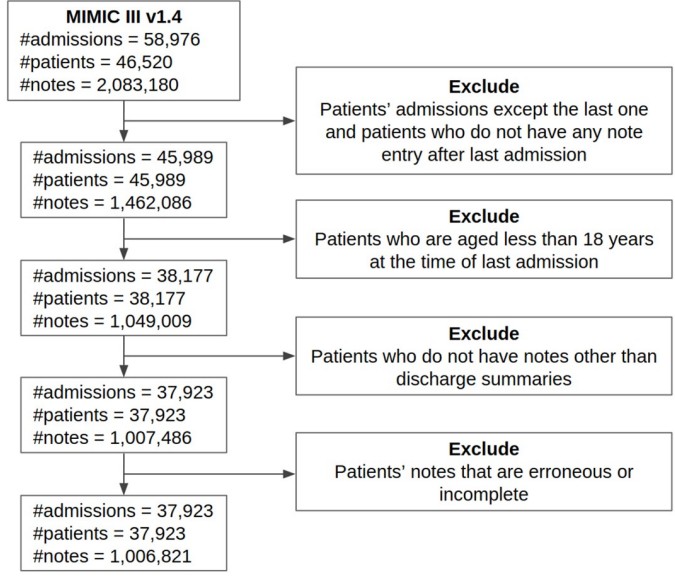

**Fig 2. Cohort selection.**

**Data processing.** We extract raw clinical notes from the TEXT field in the NOTEE-VENTS table, concatenate them per patients, and use them as model inputs. Thus, each patient will have a single note generated by concatenating all the patient's notes from the ADMIS-SIONS table. Concatenation of clinical notes reduces the problem of missing data. The outcome is the DEATHTIME field in the ADMISSIONS table. We label the death events of the patients based on the NON-NULL values in the DEATHTIME field. A NULL value in this field indicates that the patient of interest is alive.

We process the concatenated clinical notes by removing only the redacted information from the notes such as admission date, discharge date, date of birth, and age. The redacted information represents the de-identified details of the patients. We also remove repeatedly used newlines and underscores.

*Problem formulation.* We define the outcome of our mortality prediction task by the probability of being alive at a given time. Formally, for a patient, given a collection of clinical notes starting from the admission time $T_A$ to any time $T_P$, we want to predict whether, at a future time $T_F$, the patient will still be alive. Thus we formulate the mortality prediction problem using two time-windows: history window and prediction window, a terminology adopted from [28].

Fig 3 shows an illustration of the history window and the prediction window for N patients. Each patient has a variable number of clinical notes within the history window. At a given time $T_P$, for each patient, we concatenate the textual clinical notes taken within the time window from $T_A$ to $T_P$. We exclude any note recorded outside the history window to avoid data leakage. We also exclude patients who do not have any clinical notes within the designated history window and patients who passed away within the history window. To avoid redundancy, for the rest of the paper, we use the term 'note' instead of 'concatenated clinical note'.

For prediction labels, we define a time window starting from $T_P$ to $T_F$. Label 1 identifies the patients as alive while 0 means deceased at the prediction time considered.

## Feature extraction & modeling

**Feature representation.** We represent the notes using three types of feature extraction techniques: frequency-based TF-IDF, fixed embedding-based FastText, and dynamic embedding-based PubMedBERT.

*Frequency-based representation.* Frequency-based feature extraction techniques such as Term Frequency–Inverse Document Frequency (TF-IDF) [19] focus on high-frequency terms

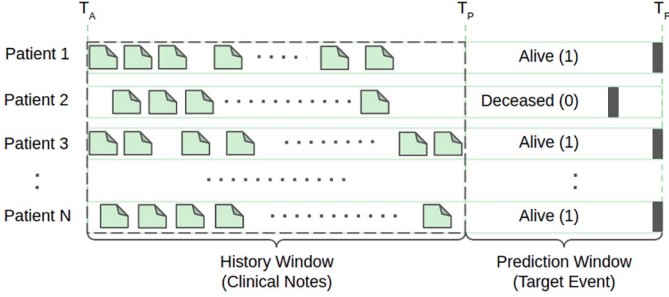

**Fig 3. Problem formulation using history window and prediction window.** $T_A$, $T_P$, and $T_F$ represent the time of admission, end of history window, and end of prediction window, respectively. The folded-corner boxes within the history window represent clinical notes from different categories such as ECG, Echo, Radiology, etc. Each patient has a variable number of clinical notes within the history window. The dark grey band within the prediction window represents the label of the patient. Label 1 means the patient is still alive, and label 0 means the patient has passed away.

in a corpus and use them to build vector representations of each document. In this work, TF-IDF transforms each vocabulary term into a single score determined by their frequency and rarity inside the clinical-notes corpus, which leads to a single vector representation of each discrete textual note.

*Fixed embedding-based representation*. Fixed embedding-based feature extraction techniques such as FastText [20], use one fixed embedding for each word irrespective of the surrounding contexts. To transform each note into vector representation by using FastText, we follow a 3-step procedure: i) tokenize the note based on white space between the words, ii) extract embeddings for each word in the note using FastText, and then ii) average over the embeddings for all words in the note to get the final vector representation of the note. Since each note in the dataset has a different number of words or phrases, the third step is required to maintain consistency over the vector representations of all notes in the dataset.

*Dynamic embedding-based representation*. Dynamic embedding-based feature representation uses dynamic word embeddings for each vocabulary word depending on the surrounding contexts. The state-of-the-art Bidirectional Encoder Representations from Transformers (BERT) [52] models use transformer-based architecture [57] to generate dynamic word embeddings. The models are pre-trained on unlabeled text data extracted from the English Wikipedia with 2,500M words and the BookCorpus [58] with 800M words [52]. There are two basic BERT models: BERT$_{Base}$ with 12 layers, 768 hidden units, and 12 heads, and BERT$_{Large}$ with 24 layers, 1024 hidden units, and 12 heads. BERT models use the WordPiece tokenization algorithm [59], and the maximum number of input tokens for BERT models can be 512.

Over recent years, text embeddings from the BERT model have been reported to outperform traditional embeddings, e.g., bag-of-words, word2vec, etc., in several general text classification tasks such as sentiment analysis [60], author attribution [61], and so on. However, in the medical domain, the original BERT model poorly performs because the clinical domain-specific natural language requires special attention for their different linguistic features and vocabulary [55].

Consequently, as our third feature extraction technique for the task, we use the PubMed-BERT encoder model [21]. PubMedBERT is pre-trained on PubMed abstracts (PubMed) and PubMed Central full-text articles (PMC). We select this model for two reasons: i) It is not trained on clinical notes from the MIMIC database. As such, PubMedBERT is not vulnerable to possible data leakage. ii) PubMedBERT has achieved state-of-the-art performance in several biomedical NLP tasks on Biomedical Language Understanding and Reasoning Benchmark [21], in comparison to BioBERT, SciBERT, CLinicalBERT. The architectural configuration of the PubMedBERT model is similar to that of the BERT$_{Base}$ model.

Using PubMedBERT for feature extraction raises a concern regarding the loss of information. This is because, PubMedBERT, like any transformer-based model, is unable to process sequences longer than 512 tokens, and as such, we need to truncate a large part of the long notes to pass them as inputs to the model. Fig 4 shows the sequence lengths of the notes after tokenization and before truncation. Here, the notes were taken within 24 hours of admission, and each note represents one patient. There are 33,740 patients with notes taken within the first day and there are 3.8 clinical notes per patient on average. As shown, in this case, if we set the sequence length to at least 2000, we will be able to cover most of the notes, and the extracted features will be less prone to information loss.

We address this issue by grouping the notes into blocks of maximum allowable sequence length, 512. We extract features from each note using PubMedBERT in three following steps:

**Step 1: Tokenizing & blocking**—We tokenize the note using PubMedBERT and then group the tokenized note into B blocks. For sequence length 512, the size of each block equals to

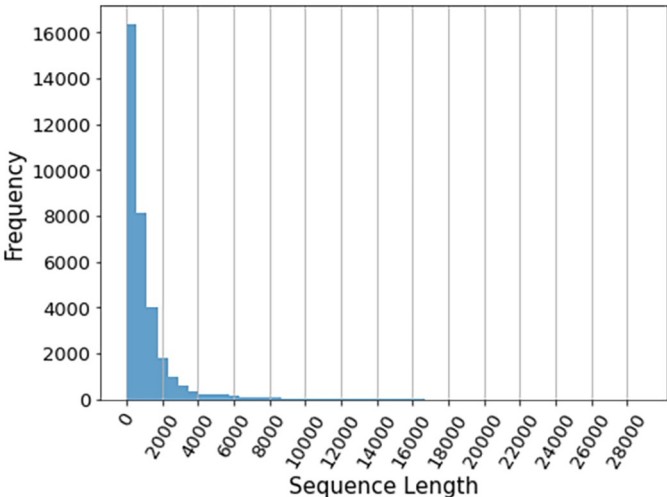

**Fig 4. Distribution of sequence length of 33,740 concatenated clinical notes after tokenization using PubMedBERT model.** The notes were taken within 24 hours after admission.

512 − 2 = 510. We spare two spaces from each block for the special tokens, [CLS] and [SEP].

**Step 2: Adding special tokens & padding**—We add [CLS] and [SEP] tokens, respectively in the beginning and at the end of each block. For cases where the length of tokenized note is not divisible by the block-size 510, we match the block-size of the last block by padding (add [PAD] token) the tokens.

**Step 3: Feature extraction**—Next, we pass the blocks of tokens to the PubMedBERT encoder. PubMedBERT has one input embedding layer and 12 encoding layers. From each of these 13 layers, we extract the embeddings of each token. We consider all layers rather than only the last one because, for general NLP tasks, each layer conveys important information on various linguistic aspects of the input text [62].

In PubMedBERT, the dimension of each token embedding is 768, and the maximum sequence length is 512. As a result, each block in each layer is represented by a tensor of size $512 \times 768$. Moreover, with B blocks per layer, the tokenized note in each layer is represented by a tensor of size $B \times 512 \times 768$. In clinical data, two notes are very unlikely to have the same number of tokens. As a result, the number of blocks is most likely to vary from one note to another, causing inconsistency among the feature representations of the notes. Thus, to maintain consistency, we average the token embeddings over B blocks in each layer and get a layer representation with 512 mean token embeddings, each with dimension 768. Note that the layer representation is defined by the features extracted from the notes by each layer.

Our ultimate goal is to extract a one-dimensional feature representation from each note. Hence, we reduce the dimension of each layer representation by averaging over 512 768-dimensional mean token embeddings and further by concatenating or averaging the mean embeddings from all 13 layers. Another way to reduce the dimensionality of the layer representations can be to use well-known dimensionality reduction techniques such as Factor Analysis, Linear Discriminant Analysis, Principal Component Analysis, Singular Value Decomposition, and so on. However, we choose to average for simplicity and leave dimensionality reduction techniques as future work.

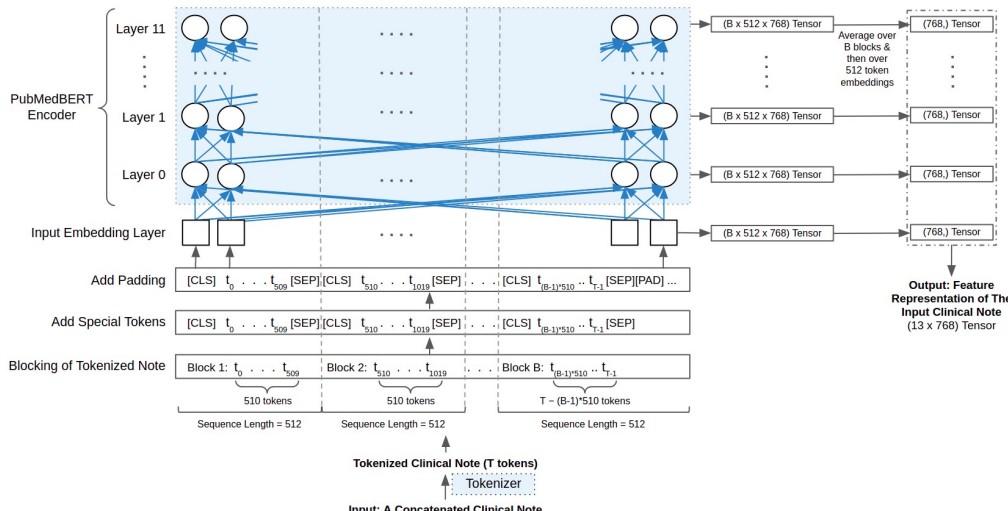

**Fig 5. Illustration of feature extraction from one note using the PubMedBERT model.** Here *B* refers to the number of blocks, $t_i$ refers to the i-th token, and *T* refers to the total number of tokens.

Fig 5 illustrates the steps involved in feature extraction from the notes using the PubMed-BERT model. Here *B* refers to the number of blocks, $t_i$ refers to the i-th token, and *T* refers to the total number of tokens. The blue boxes represent the PubMedBERT tokenizer and the PubMedBERT model. For *T* tokens in a concatenated clinical note, we have $B = T/510$ blocks, and the value of *B* varies from one note to another.

**Prediction models.** For our prediction task, we use four machine learning classification models: logistic regression, light gradient boosting machine (Light-GBM), random forest, and a 3-layer feed-forward neural network (FFNN).

**Metrics.** For evaluating the performance of our prediction models and the feature representations, we use three metrics: area under the receiver operator characteristics curve (AU-ROC) score, specificity score, and sensitivity score. AU-ROC tells us how well the classification models can distinguish between two classes. The outcome of our mortality prediction task is the probability of being alive. Thus, the sensitivity score defines how well the classification models can identify patients who will still be alive at a given future time and the specificity score defines the models' ability to identify patients who will be deceased by that time. We calculate the sensitivity and specificity scores by using optimal threshold value calculated by Youden's index [63]:

$$J = sensitivity + specificity - 1 \qquad (1)$$

Note that area under the precision-recall curve (AU-PRC) is also a widely used evaluation metric in mortality prediction task. However, since AU-PRC focuses on positive class—alive, and our dataset is highly skewed towards alive patients, it is expected to bear the least significance in our mortality prediction task. As such we do not consider AU-PRC as one of the evaluation metrics in this work.

## Experimental setup

We set up a three-part experiment for the ICU mortality prediction task: i) short-term prediction, ii) mid-term prediction, and iii) long-term prediction. We choose the short-term, mid-term, and long-term prediction window by considering the the distribution of the in-hospital

length of stay (LOS) of the patients in our selected cohort. We calculate the LOS for each patient by using the time of last admission from ADMITTIME and the time of discharge from DISCHTIME fields in the ADMISSIONS table. In our selected cohort, 25% of the patients have a 4-day LOS, 50% of the patients have a 7-day LOS, and 75% of the patients have a 12-day period LOS. The average LOS is 10 days and the standard deviation is 10 days.

We refer to the 48-hour and 4-day period as short-term, 7-day and 10-day period for mid-term, 15-day and 30-day period for long-term prediction window. For the history window, we always choose the first 24 hours after admission. There are 33,740 patients who have clinical notes within the first 24 hours after admission.

**Short-term prediction.** As described in section *Data processing*, we label the patients as either deceased (0) or alive (1) based on the prediction window. As such, at the end of the prediction window, we have 33,238 alive patients and 502 deceased patients. We then create stratified 80-20 train-test splits of patients' notes. We use the train set for training the predictive models, and the test set for reporting the performance metrics.

We extract features from the notes using TF-IDF, FastText, and PubMedBERT. In TF-IDF implementation using scikit-learn, we set *ngram_range* = (1, 2), which implies that during feature extraction, TF-IDF will consider words (unigrams) as well as phrases (bigrams). We find the optimal size of TF-IDF vector representation by experimenting with the *max_features* parameter. We consider three variants of the maximum features, 1000, 5000, 10000, 50000 and choose the best one depending on the AU-ROC score. For FastText, on the other hand, we fine-tune the pre-trained FastText model [20] on clinical notes in the train set and choose the suitable dimension for feature representation by experimenting with 50 and 300. We choose these two dimensions based on the original FastText paper [20]. Last but not least, for the pre-trained PubMedBERT model, we consider two types of feature representations as well: concatenated embeddings from all 13 layers with $13^*768 = 9,984$ dimension, and mean embeddings by averaging over all layers with 768 dimension. We then pick the best one depending on the AU-ROC score. We implement the pre-trained PubMedBERT model and tokenizer using the huggingface API [64].

Note that our dataset is highly skewed towards alive patients (explained later in section *Mid-term & long-term prediction*). We resolve this class imbalance by setting the *class_weight* parameter as "*balanced*" in logistic regression, Light-GBM, and random forest implementations [16, 65]. In the implementation of the feed-forward neural network with PyTorch [66], we calculate the class weights manually and use them to weigh the training loss. Note that the concatenating of notes ensures avoiding the risk of data leakage in the test set. It is because, after concatenation, there is only one entry for each patient in the dataset.

We consider the cross-entropy loss function, and adam optimizer [67] for our feed-forward neural network. For each of the four machine learning models, we perform hyperparameter optimization to find the best suited hyperparameters for this task. To reduce computational cost, we optimize the hyperparameters from all four models on our short-term (2-day) mortality prediction task. Additionally, for feature representation, we use TF-IDF with 1000 maximum features, FastText with 50 dimension, and PubMedBERT with 784 features. We consider the following combinations of hyperparameters: C = [0.001, 0.5, 5] for logistic regression; num_leaves = [2, 10, 100], max_depth = [2, 10, 50], learning_rate = [0.01, 0.1, 0.5], n_estimators = [80, 100, 500], min_child_weight = [0.001, 0.1, 0.5], min_child_samples = [2, 20, 50] for light-GBM; n_estimator = [50, 100, 500], max_depth = [None, 10, 50], max_features = ['auto', 'sqrt', 'log2'], min_samples_split = [0.5, 2, 10], min_samples_leaf = [0.1, 1, 4] for random forest; learning_rate = [0.02, 0.1], hidden_nodes = [50, 100, 200] for feed-forward neural network. The description of each of these parameters can be found in the Scikit-Learn, LightGBM, and PyTorch documentations. We choose the most suited combination of hyperparameters for

**Table 1. Class distribution for short/mid/long-term prediction window.**

| Prediction Task | Prediction Window (in days) | #Deceased Patients | #Alive Patients | Deceased to Alive Ratio |
|---|---|---|---|---|
| Short-term | $1 < t \leq 2$ | 502 | 33,238 | ∼1:66 |
|  | $1 < t \leq 4$ | 1,197 | 32,543 | ∼1:27 |
| Mid-term | $1 < t \leq 7$ | 2,044 | 31,696 | ∼1:16 |
|  | $1 < t \leq 10$ | 2,586 | 31,154 | ∼1:12 |
| Long-term | $1 < t \leq 15$ | 3,175 | 30,565 | ∼1:10 |
|  | $1 < t \leq 30$ | 3,877 | 29,863 | ∼1:8 |

each of the four models depending on the highest AU-ROC scores and further our experiments.

**Mid-term & long-term prediction.** Table 1 shows the class distribution for the short/mid/long-term prediction tasks. As we increase the length of the prediction window, the number of alive patients per deceased patient decreases. Note that the total number of patients for all prediction windows is equal. This is because, at all times, we have the same cohort of patients.

For the mid-term and long-term mortality prediction tasks, we pick the three best feature representations from each of the three feature extraction techniques: TF-IDF, FastText, and PubMedBERT based on the experiments performed in section *Short-term prediction*. Furthermore, for predictive modeling, we pick the best-performing model from the short-term prediction task.

## Results & discussion

In this section, we report and discuss the experimental results and their implications.

### Hyperparameter optimization

Based on the highest AU-ROC scores, in Table 2, we show the best combinations of hyperparameters for each model.

**Table 2. Hyperparameter optimization for classification models.**

| Classification Model | Hyperparameters | TF-IDF | FastText | PubMedBERT |
|---|---|---|---|---|
| Logistic Regression | C | 0.5 | 0.5 | 0.5 |
| Light-GBM | num_leaves | 2 | 100 | 100 |
|  | max_depth | [2, 10, 50] | 50 | 50 |
|  | learning_rate | 0.1 | 0.01 | 0.1 |
|  | n_estimators | 500 | 500 | 500 |
|  | min_child_weight | [0.001, 0.1, 0.5] | [0.001, 0.1, 0.5] | 0.5 |
|  | min_child_samples | 50 | 50 | 50 |
| Random Forest | n_estimator | 500 | 500 | 500 |
|  | max_depth | None | None | None |
|  | max_features | ['auto', 'sqrt'] | 'auto' | 'auto' |
|  | min_samples_split | 2 | 10 | 10 |
|  | min_samples_leaf | 1 | 1 | 4 |
| Feed-forward Neural Network | learning_rate | 0.02 | 0.02 | 0.02 |
|  | hidden_nodes | 50 | 50 | 50 |

As shown, for TF-IDF, the best AU-ROC scores are given by the following combinations: C = 0.5 for logistic regression; num_leaves = 2, max_depth = [2, 10, 50], learning_rate = 0.1, n_estimators = 500, min_child_weight = [0.001, 0.1, 0.5], min_child_samples = 50 for light-GBM; n_estimator = 500, max_depth = None, max_features = ['auto', 'sqrt'], min_samples_split = 2, min_samples_leaf = 1 for random forest; learning_rate = 0.02, hidden_nodes = 200 for feed-forward neural network.

for FastText, the best AU-ROC scores are given by the following combinations: C = 0.5 for logistic regression; num_leaves = 100, max_depth = 50, learning_rate = 0.01, n_estimators = 500, min_child_weight = [0.001, 0.1, 0.5], min_child_samples = 50 for light-GBM; n_estimator = 500, max_depth = None, max_features = 'auto', min_samples_split = 10, min_samples_leaf = 1 for random forest; learning_rate = 0.02, hidden_nodes = 200 for feed-forward neural network.

for PubMedBERT, the best AU-ROC scores are given by the following combinations: C = 0.5 for logistic regression; num_leaves = 100, max_depth = 50, learning_rate = 0.1, n_estimators = 500, min_child_weight = 0.5, min_child_samples = 50 for light-GBM; n_estimator = 500, max_depth = None, max_features = 'auto', min_samples_split = 10, min_samples_leaf = 4 for random forest; learning_rate = 0.02, hidden_nodes = 200 for feed-forward neural network.

When multiple values of a certain hyperparameter help the model achieve the highest AU-ROC score, such as max_depth = [2, 10, 50] for light-GBM, we randomly choose the first value in the list as the optimal one.

## Short-term prediction

We report the AU-ROC scores achieved for the test set by the four models, trained with eight variants of feature representations. We extract each of the representations from notes taken within the first 24 hours of admission. We provide a 2-fold discussion of the results presented in Tables 3–5: model-wise and feature representation-wise.

As shown, the logistic regression model consistently outperformed light-GBM, random forest, and feed-forward neural network, irrespective of the different feature representations. It indicates that for our mortality prediction task, a simpler linear model such as logistic regression is more suitable with high AU-ROC scores 0.86 and 0.83 for 48-hour and 4-day mortality prediction respectively. Moreover, logistic regression is faster to train and more interpretable.

Furthermore, TF-IDF feature representation (Table 3), with 50,000 maximum features, performed the best with logistic regression. As known, in TF-IDF, an increasing number of

**Table 3. Test AU-ROC scores by four models trained with features extracted using TF-IDF.**

| Prediction Window | Dimension of Feature Representations | Logistic Regression | Light-GBM | Random Forest | Feed-forward Neural Network |
|---|---|---|---|---|---|
| 1 < t ≤ 2 | 1,000 | 0.839 | 0.831 | 0.788 | 0.83 |
| | 5,000 | 0.864 | 0.841 | 0.829 | 0.843 |
| | 10,000 | 0.867 | 0.840 | 0.829 | 0.836 |
| | 50,000 | **0.872** | 0.842 | 0.816 | 0.787 |
| 1 < t ≤ 4 | 1,000 | 0.808 | 0.791 | 0.779 | 0.77 |
| | 5,000 | 0.829 | 0.808 | 0.790 | 0.816 |
| | 10,000 | 0.830 | 0.815 | 0.797 | 0.821 |
| | 50,000 | **0.832** | 0.809 | 0.793 | 0.796 |

The columns show the comparison among the models, and the rows show the comparison among three variants of maximum features in TF-IDF. The bold numbers indicate the highest AU-ROC scores and the best performing models for both short-term prediction windows.

**Table 4. Test AU-ROC scores for four models trained with features extracted using FastText.**

| Prediction Window | Dimension of Feature Representations | Logistic Regression | Light-GBM | Random Forest | Feed-forward Neural Network |
|---|---|---|---|---|---|
| $1 < t \leq 2$ | 50 | 0.807 | 0.805 | 0.785 | 0.678 |
| | 300 | **0.832** | 0.801 | 0.775 | 0.649 |
| $1 < t \leq 4$ | 50 | 0.74 | 0.744 | 0.743 | 0.629 |
| | 300 | **0.771** | 0.758 | 0.747 | 0.641 |

The columns show the comparison among the models, and the rows show the comparison among two variants of dimensions in FastText features. The bold numbers indicate the highest AU-ROC scores and the best performing models for both short-term prediction windows.

maximum features implies the addition of vocabulary in the feature representation. As such, adding more vocabulary in the feature representation of each note captures more information about the state of the patient and is beneficial to their mortality prediction. However, The difference between the AU-ROC scores for 5000, 10000, and 50000 being very low ($\sim$ 0.003-0.008) and considering the computational cost of classification for high dimensional feature representations, we pick 5000 TF-IDF maximum features for further experiments. Thus, the best achieved AU-ROC score is 0.864.

For FastText (Table 4), logistic regression performed the best with 300-dimensional feature representation. The highest achieved AU-ROC is 0.832. In FastText, each dimension in the feature representation captures the morphological meaning of the words in the notes. That being said, with 300 dimensions, more information about the words are captured, and as such are contributing towards the better performance of the classification model.

Last but not least, for the PubMedBERT model (Table 5), we notice a higher AU-ROC score (0.837) achieved by the logistic regression model with concatenated feature representations than with mean feature representations. It implies that each layer in the PubMedBERT model captures information about the patients' mortality from the notes. As we concatenate the feature representations from all 13 layers of the model, the information from each layer is kept separate from one another and later used by the classification model.

Later, we provide more details on the high-importance features from each of these feature representations in section *Interpretability of predictive model*.

## Mid-term & long-term prediction

Based on the highest AU-ROC scores from the short-term mortality prediction, for the mid and long-term prediction tasks, we choose the logistic regression model with three feature representations: features from TF-IDF with 5,000 maximum features, from FastText with 300 dimensions, and from PubMedBERT with 9,984-dimensions. Choosing only one classification model also reduces the runtime for the prediction tasks.

**Table 5. Test AU-ROC scores for four models trained with features extracted using PubMedBERT.**

| Prediction Window | Dimension of Feature Representations | Logistic Regression | Light-GBM | Random Forest | Feed-forward Neural Network |
|---|---|---|---|---|---|
| $1 < t \leq 2$ | 784(mean) | 0.835 | 0.808 | 0.769 | 0.64 |
| | 9,984(concatenated) | **0.837** | 0.832 | 0.77 | 0.636 |
| $1 < t \leq 4$ | 784(mean) | 0.768 | 0.752 | 0.745 | 0.602 |
| | 9,984(concatenated) | **0.77** | 0.767 | 0.756 | 0.59 |

The columns show the comparison among the models, and the rows show the comparison among two variants of dimensions in PubMedBERT features. The bold numbers indicate the highest AU-ROC scores and the best performing models for both short-term prediction windows.

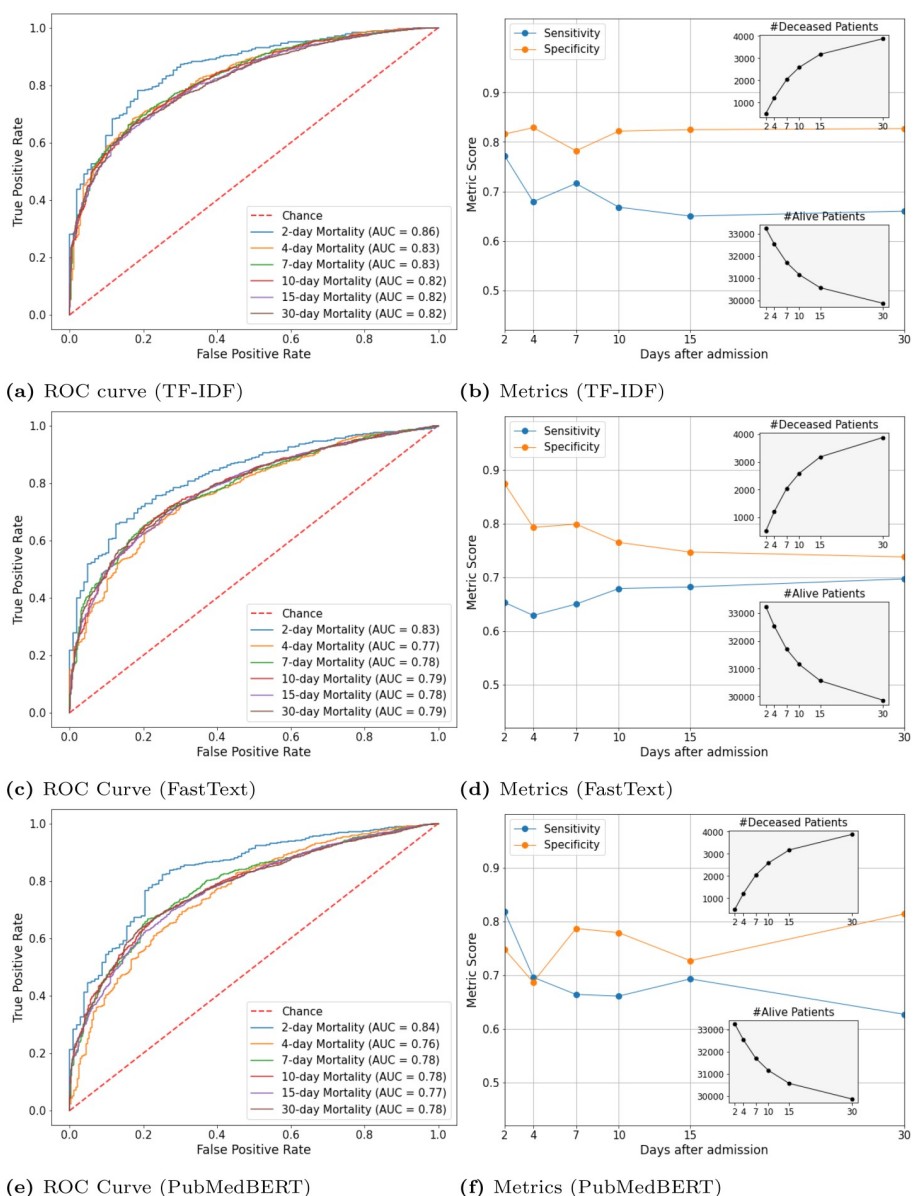

**Fig 6. Test ROC curve, AU-ROC score, sensitivity and specificity scores for logistic regression model with TF-IDF, FastText, and PubMedBERT.** The optimum threshold value for sensitivity and specificity scores has been calculated using Youden's Index. The x-axis represents the prediction window. The grey boxes show the number of deceased and alive patients with respect to the prediction windows.

Fig 6 shows the ROC curves along with the AU-ROC scores (Fig 6a, 6c and 6e) and sensitivity and specificity scores along with the number of deceased and alive patients (Fig 6b, 6d and 6f) for 2-day, 4-day, 7-day, 10-day, 15-day, and 30-day mortality prediction.

As shown in Fig 6a, 6c and 6e, the highest AU-ROC scores, 0.86 with TF-IDF, 0.83 with FastText, and 0.84 with PubMedBERT were achieved by the short-term (48-hour prediction window after admission) prediction task.

Interestingly, with the increasing number of days after admission and with the increasing number of deceased patients, we notice similar AU-ROC scores: $\sim$ 0.82 with TF-IDF, $\sim$ 0.78

with FastText, and ∼ 0.77 with PubMedBERT. This indicates that given only 24 hours of patients' notes, the logistic regression model can identify deceased or alive patients over a long period (at least up to 30 days). Furthermore, the sensitivity and the specificity scores in Fig 6b, 6d and 6f reflect that even with the increasing number of days since admission, the model has been able to maintain consistently high specificity scores: > 0.75 with TF-IDF, > 0.7 with both FastText and PubMedBERT. Recall that the specificity scores indicate the model's ability to correctly identify the patients who will not survive at the end of the prediction window. We highly emphasize the specificity because domain experts are likely to be more cautious about correctly predicting dying patients, i.e., high specificity. Mispredicting a dying patient will have a catastrophic impact on the critical medical decision-making process. As such, maintaining high specificity scores is vital to save patients.

Furthermore, we notice that the logistic regression classifier with TF-IDF feature representation shows consistently high sensitivity scores, approximately 0.7. It implies that the model can identify alive patients even if we move forward in time.

## Interpretability of predictive model

In this section, we interpret what the logistic regression model has learned during training. We do this by considering the importance scores and the statistical significance of the features learned by the logistic regression model. We calculate the feature importance scores from the coefficient of the features in the decision function of logistic regression, our best performing model. A higher positive coefficient value indicates higher influence in predicting the probability of being alive and a lower negative coefficient indicates higher influence in predicting the probability of being deceased. We also measure the statistical significance of the features by calculating their p-values. We define the null hypothesis as follows: *there is no significant relationship between the predictors (features) and predicted variables (deceased or alive) in the mortality prediction tasks*. We consider the confidence level to be 0.01; i.e., for each feature, we reject the null hypothesis if the p-value is less than 0.01.

**Interpretation of feature-significance.** We hypothesize that the extracted features from the clinical notes taken at a certain time period should have lesser significance in predicting mortality as we move forward in time. Interestingly, for the TF-IDF feature representations, the number of significant features for 2-day, 4-day, 7-day, 10-day, 15-day, and 30-day mortality prediction tasks is 620, 436, 297, 264, 233, and 187, respectively. Recall that, TF-IDF features represent the terms (words/phrases) within the clinical notes. Thus, the number of significant terms decreases with the increasing length of the prediction window, which aligns with our aforementioned hypothesis.

**Interpretation of the significant features.** We provide a 2-fold discussion on the interpretability of the significant features learned by the classification model. First, we show (Fig 7) the top 10 most important and statistically significant features (terms) for predicting the probability of being alive and the top 10 most important and statistically significant terms for predicting the probability of being deceased.

Then, we show (Fig 8) a detailed view of the significant terms that are common over the six prediction windows along with their importance scores.

The red bars in Fig 7 show the top 10 most important and statistically significant terms for predicting the probability of being deceased. We notice that the model is providing more focus on words such as 'herniation', 'unresponsive', 'tachycardia', 'metastatic', 'infarct', 'worsening', 'hemorrhage', 'irregular' and so on. On the other hand, to predict the probability of being alive (green bars in Fig 7), the model is providing more focus on the terms such as 'sinus bradycardia', 'small', 'normal', 'modest', 'weaned', 'unremarkable' and so on.

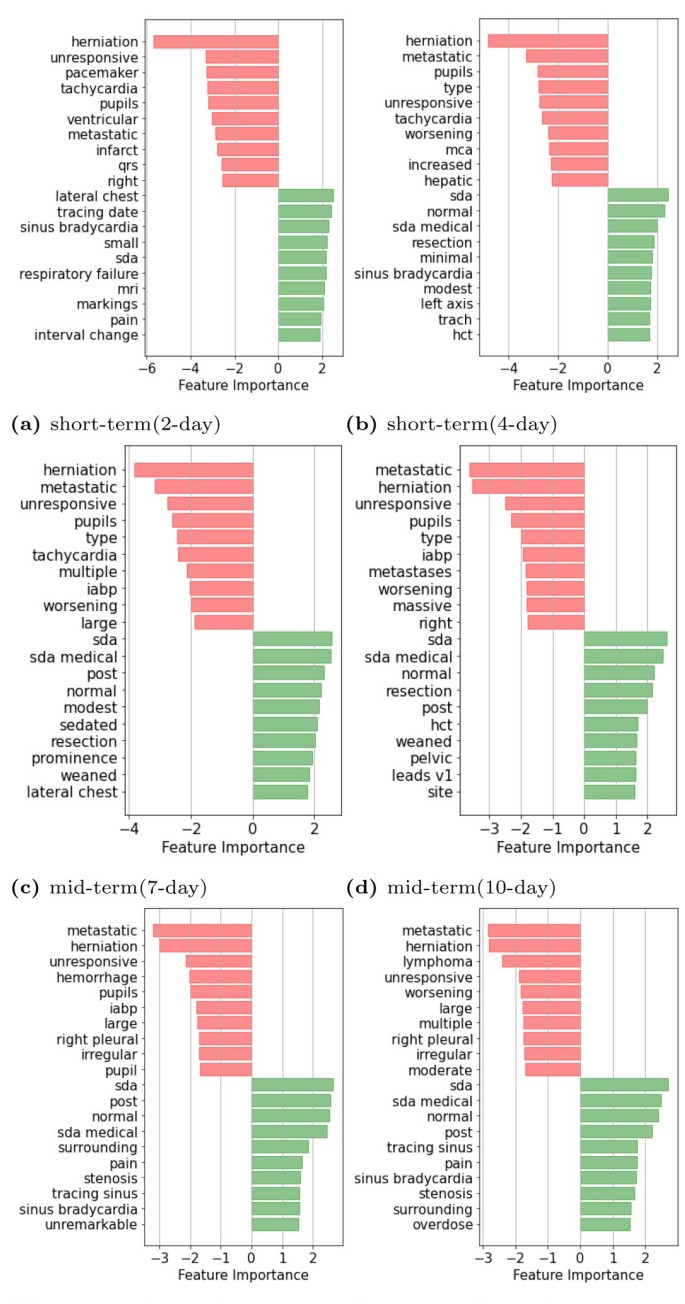

**Fig 7. Top 10 most important features that are predictive of the mortality prediction outcome—Alive (green bars), deceased (red bars), by logistic regression model with TF-IDF.**

There are 80 features that remained significant for all short/mid/long-term mortality prediction tasks. Fig 8 shows the significant terms that are common for short(2-day)/mid (10-day)/long-term(30-day) mortality prediction tasks. More specifically, Fig 8a and 8b show the feature importance of the terms that are given more importance in predicting the probability of being deceased and alive, respectively. As shown in Fig 8a, the classification model provides more emphasize on terms 'sob reason', 'metastatic', 'interstitial', 'pupil', 'increased', as it

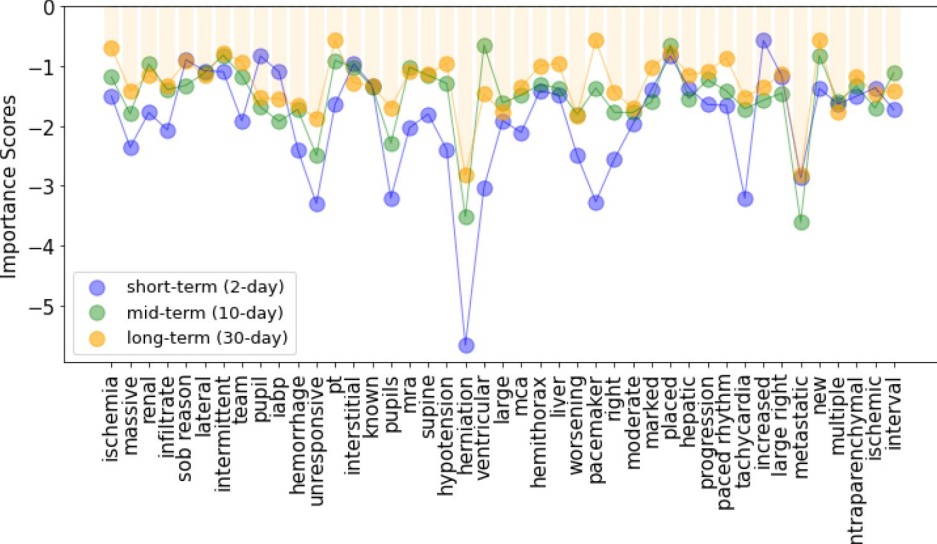

**(a)** Features assisting to predict the probability of being deceased

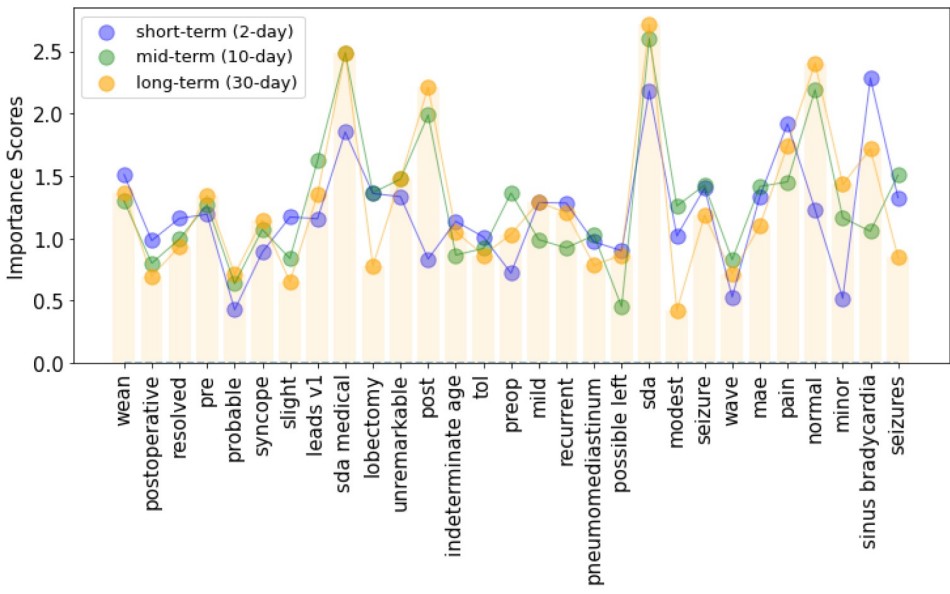

**(b)** Features assisting to predict the probability of being alive

**Fig 8. Importance of the features that remained significant over short/mid/long-term prediction windows.** These features are from the logistic regression with TF-IDF.

attempts to predict the probability of being deceased in the long run (mid/long-term mortality prediction). In contrast, Fig 8b shows that to predict the mid-term and long-term probability of being alive, the model puts more focus on terms such as 'normal', 'minor', 'sda medical', and 'post'. We hope that our attempt to interpret the machine learning model using the above mentioned association of the importance scores of different terms with different prediction windows will help the domain experts to understand the model's learning paradigm.

The feature importance scores given by the logistic regression models with FastText and PubMedBERT provide an insight into the roles played by each of the feature dimensions.

However, since they do not explicitly reveal the dominant terms in the clinical notes, we do not include the feature importance scores given by the logistic regression models with FastText and PubMedBERT in this discussion.

## Short-term vs. mid-term vs. long-term mortality

According to the results of our experiments, the best performing model, logistic regression achieved the following AU-ROC scores: 0.86 for short-term (2-day) mortality, 0.83 for short-term (4-day) and mid-term (7-day) mortality, and 0.82 for mid-term (10-day) and long-term mortality. The drop in AU-ROC scores can be explained by the 24-hour history window. Because we use the clinical notes taken within only 24 hours after admission, many of the features (terms) in the notes bear lesser significance as we try to forecast patients' mortality in the long run. Interestingly, the AU-ROC scores for the mid-term and long-term mortality prediction tasks do not drop more than 0.04.

Since, from section *Mid-term & long-term prediction*, we observe that our data is highly skewed towards the alive patient samples, we tackle the class imbalance issue (described in section *Short-term prediction*) by weighing the training loss of the model. Intuitively, when we multiply the loss with high weight for the deceased-patient class, the optimizer takes bigger steps towards the global minimum of the loss function while updating the model parameters during the training phase. A bigger step can result in either of the two updates: i) ideal updates where the optimizer takes steps towards the global minimum, ii) noisy updates where the optimizer takes steps away from the global minimum. Noisy updates are most likely to result in lower than the expected performance of the classification models, and practically, in most real-world scenarios, we observe noisy updates with bigger steps. Recall from section *Mid-term & long-term prediction*, when we increase the prediction window, the weight for the deceased-patient class decreases. Thus with lower weights, we expect the optimizer to take smaller steps and perform less noisy parameter updates during mid/long-term mortality prediction tasks, resulting in higher than expected performance of the classification model.

## TF-IDF vs. FastText vs. PubMedBERT

We use three kinds of feature extraction techniques, frequency-based TF-IDF, fixed embedding-based FastText, and dynamic embedding-based PubMedBERT throughout the experiments, Each of them uses a different perspective to extract features from the notes. While TF-IDF used frequency of word appearances, FastText used a morphological representation of the words. On the other hand, the state-of-the-art PubMedBERT model used the surrounding contexts of the words. In section *Interpretability of predictive model*, we have shown how the features from these techniques contribute to the model's learning paradigm.

However, as the results from section *Short-term prediction*, and *Mid-term & long-term prediction* suggest, among the three techniques, TF-IDF exhibits the most promising performance in this setting of our mortality prediction task. It may happen due to the simplicity in the mechanism of this feature extraction technique. Nonetheless, to conclude the certainty of which one is the best, we need to further experiment with more varieties of feature representations. Some of them can be as follows: i) in FastText, instead of averaging the word embeddings, considering a certain number of words and concatenating them to represent the notes. However, this would require an extensive study of the words to establish a selection criteria so that there is no possibility for loss of information. ii) in PubMedBERT, considering well-established dimensionality reduction techniques to reduce the dimension of the features rather than using simple average, or using a smaller sequence length, and so on.

As the focus of our study is to predict short/mid/long-term mortality in ICU patients by using only free-text clinical notes from 24-hours after admission and simple interpretable machine learning algorithms, we focus only on the task in this paper and leave model/feature improvement for future work.

## Conclusion & future work

Mortality prediction of ICU patients is a fundamental problem in the domain of medical informatics. In this work, we have presented a framework for predicting short-term, mid-term, and long-term probability of being alive in adult ICU patients by using unstructured clinical notes and a cohort from the MIMIC III database. We have experimented with four machine learning algorithms and three kinds of feature representations. We have shown that by only using clinical notes taken within the first 24 hours after admission, the best performing model can achieve high AU-ROC scores: 0.86 for short-term (2-day) mortality, 0.83 for short-term (4-day) and mid-term (7-day) mortality, and 0.82 for mid-term (10-day) and long-term mortality. We have compared three different types of feature extraction techniques and found that despite introducing state-of-the-art models like PubMedBERT and FastText, TF-IDF has the best AU-ROC score for this task. This study is the first of its kind to compare three types of feature representations extracted from clinical notes using three different feature extraction techniques from NLP: frequency-based technique, fixed embedding-based technique, and dynamic embedding-based technique. To serve as a basis for the interpretability of our predictive model, we have also presented the top features extracted from the clinical notes by TF-IDF feature extraction techniques. Overall, this paper emphasizes the integration of NLP in clinical outcome prediction tasks and explores the immense potential of raw clinical notes.

Some of the future research directions that can stem from this work are as follows: i) Using a variety of FastText and PubMedBERT feature representations other than the ones used in this paper, some of which can be using concatenated feature representation from FastText, considering well-known dimensionality reduction techniques for features extracted by PubMedBERT, considering different sequence lengths in PubMedBERT, and so on. ii) Examining the robustness and resiliency of the feature representations and the predictive models against vulnerabilities of the raw clinical notes such as missingness, incoherence, spelling mistakes, etc. iii) Using only structured data as well as both structured data and unstructured clinical notes for comparative analysis.

## Author Contributions

**Conceptualization:** Maria Mahbub, Sudarshan Srinivasan, Ioana Danciu.

**Data curation:** Maria Mahbub.

**Formal analysis:** Maria Mahbub, Sudarshan Srinivasan, Ioana Danciu, Alina Peluso.

**Funding acquisition:** Edmon Begoli.

**Investigation:** Maria Mahbub.

**Methodology:** Maria Mahbub, Sudarshan Srinivasan.

**Software:** Maria Mahbub.

**Supervision:** Edmon Begoli, Gregory D. Peterson.

**Validation:** Maria Mahbub.

**Visualization:** Maria Mahbub.

**Writing – original draft:** Maria Mahbub.

**Writing – review & editing:** Maria Mahbub, Sudarshan Srinivasan, Ioana Danciu, Alina Peluso, Edmon Begoli, Suzanne Tamang, Gregory D. Peterson.

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
