## [Decision Letter · Decision Letter 0]

10 Dec 2021

PONE-D-21-29049Unstructured clinical notes within the 24 hours since admission predict short, mid & long-term mortality in adult ICU patientsPLOS ONE

Dear Dr. Maria Mehboob,

Thank you for submitting your manuscript to PLOS ONE. After careful consideration, we feel that it has merit but does not fully meet PLOS ONE’s publication criteria as it currently stands. Therefore, we invite you to submit a revised version of the manuscript that addresses the points raised during the review process.

We look forward to receiving your revised manuscript.

Kind regards,

Wajid Mumtaz

Academic Editor

PLOS ONE

“This research used resources of the Knowledge Discovery Infrastructure at the Oak Ridge National Laboratory, which is supported by the Office of Science of the U.S. Department of Energy under Contract No. DE-AC05-00OR22725 and the Department of Veterans Affairs Office of Information Technology Inter-Agency Agreement with the Department of Energy under IAA No. VA118-16-M-1062”

“This research used resources of the Knowledge Discovery Infrastructure at the Oak 598 Ridge National Laboratory, which is supported by the Office of Science of the U.S. 599 Department of Energy under Contract No. DE-AC05-00OR22725 and the Department 600 of Veterans Affairs Office of Information Technology Inter-Agency Agreement with the 601 Department of Energy under IAA No. VA118-16-M-1062”

“This research used resources of the Knowledge Discovery Infrastructure at the Oak Ridge National Laboratory, which is supported by the Office of Science of the U.S. Department of Energy under Contract No. DE-AC05-00OR22725 and the Department of Veterans Affairs Office of Information Technology Inter-Agency Agreement with the Department of Energy under IAA No. VA118-16-M-1062”

Reviewers' comments:

Reviewer's Responses to Questions

**Comments to the Author**

1. Is the manuscript technically sound, and do the data support the conclusions?

Reviewer #1: Yes

2. Has the statistical analysis been performed appropriately and rigorously? 

Reviewer #1: Yes

3. Have the authors made all data underlying the findings in their manuscript fully available?

Reviewer #1: Yes

4. Is the manuscript presented in an intelligible fashion and written in standard English?

Reviewer #1: Yes

5. Review Comments to the Author

Reviewer #1: The authors performed a retrospective analysis of MIMIC data using NLP to parse clinical notes written within 24 hours of admission in predicting mortality at several intervals (48 hours, 4 days, 7 days, 10 days, 15 days, and 30 days) with four different machine learning approaches (LR, GBM, RF, neural network), achieving strong discrimination using unstructured data alone. The authors also investigate three different NLP feature extraction techniques and explore interpretability mechanisms (feature importance). The topic is important. The methods are reasonable. The time horizons are appropriate.

Why not use both unstructured and structured clinical data? As the authors note, their is some overlap between these entities, and also some unique information that is contained only in unstructured data and some that is contained only in unstructured data. Unstructured data has a theoretical advantage of greater objectivity. As you may know, a group from Boston and industry have shared the perspective that machine learning applications often learn clinicians' thought processes even when using structured data, producing outputs that tell the clinician that which they already knew (PMID: 33785839). Unstructured data from clinical notes seems to reflect clinicians' thought process to an even greater degree. Please discuss these perspectives and justify the use of clinical notes alone in prediction modeling.

6. PLOS authors have the option to publish the peer review history of their article (what does this mean?). If published, this will include your full peer review and any attached files.

Reviewer #1: **Yes: **Tyler Loftus

---

## [Author Response · Author response to Decision Letter 0]

14 Dec 2021

Reviewer Points:

- Why not use both unstructured and structured clinical data? As the authors note, their is some overlap between these entities, and also some unique information that is contained only in unstructured data and some that is contained only in unstructured data. Unstructured data has a theoretical advantage of greater objectivity. As you may know, a group from Boston and industry have shared the perspective that machine learning applications often learn clinicians' thought processes even when using structured data, producing outputs that tell the clinician that which they already knew (PMID: 33785839). Unstructured data from clinical notes seems to reflect clinicians' thought process to an even greater degree. Please discuss these perspectives and justify the use of clinical notes alone in prediction modeling.

Answer: We agree with the reviewer. As the unstructured data occupy 80% of the EHR, they can capture more detailed information on patients, resulting in better performance of machine learning models. As such, models trained on these data may better capture the thought process of the clinicians and the conditions of the patients compared to models trained on structured data. However, to conclude the advantages of using unstructured clinical notes over structured data for predictive modeling in our problem setting, we need to further experiment with only structured data as well as both structured and unstructured data. But, since the focus of this study is to unveil the potential of raw unstructured clinical notes for predicting mortality in adult ICU patients, we use solely unstructured clinical notes in prediction modeling for this work and leave the comparative analysis for future work.

We added a brief discussion on these perspectives of using only clinical notes in prediction modeling in the Introduction section of the paper.

Editorial comments:

Answer: We revised the manuscript to meet the style requirements of the journal, including section headings, figure captions and references, and table captions and references.

2. Please state what role the funders took in the study. Please include this amended Role of Funder statement in your cover letter; we will change the online submission form on your behalf.

Answer: We have added the role of funders in the study. Here is the amended Role of Funder statement: “The funders had no role in study design, data collection and analysis, decision to publish, or preparation of the manuscript.”

3. Please remove any funding-related text from the manuscript and let us know how you would like to update your Funding Statement. Please include your amended statements within your cover letter; we will change the online submission form on your behalf.

Answer: We have removed all funding-related texts from the manuscript. We have removed the acknowledgment section from the manuscript and would like to update the Funding statement in the online submission form. Here is the amended statement: “This research used resources of the Knowledge Discovery Infrastructure at the Oak Ridge National Laboratory, which is supported by the Office of Science of the U.S. Department of Energy under Contract No. DE-AC05-00OR22725 and the Department of Veterans Affairs Office of Information Technology Inter-Agency Agreement with the Department of Energy under IAA No. VA118-16-M-1062. This manuscript has been in part co-authored by UT-Battelle, LLC under Contract No. DE-AC05-00OR22725 with the U.S. Department of Energy. The United States Government retains and the publisher, by accepting the article for publication, acknowledges that the United States Government retains a non-exclusive, paid-up, irrevocable, world-wide license to publish or reproduce the published form of this manuscript, or allow others to do so, for United States Government purposes. The Department of Energy will provide public access to these results of federally sponsored research in accordance with the DOE Public Access Plan (http://energy.gov/downloads/doe-public-access-plan). The funders had no role in study design, data collection and analysis, decision to publish, or preparation of the manuscript. The specific roles of all authors are articulated in the ‘author contributions’ section.”

4. PLOS requires an ORCID iD for the corresponding author in Editorial Manager on papers submitted after December 6th, 2016. Please ensure that you have an ORCID iD and that it is validated in Editorial Manager. 

Answer: We have added and validated the ORCID iD for the corresponding author in the Editorial Manager.

Answer: We have edited the ethics statement in the Methods section of our manuscript and removed it from the other section(s).

Answer: We have reviewed the reference list to check for quality. We have updated several references to be more complete.

7. While revising your submission, please upload your figure files to the Preflight Analysis and Conversion Engine (PACE) digital diagnostic tool.

Answer: We have used PACE to make appropriate corrections to the figure files. We have uploaded the corrected figure files.

---

## [Decision Letter · Decision Letter 1]

19 Dec 2021

Unstructured clinical notes within the 24 hours since admission predict short, mid & long-term mortality in adult ICU patients

PONE-D-21-29049R1

Dear Dr. Maria Mahbub,

We’re pleased to inform you that your manuscript has been judged scientifically suitable for publication and will be formally accepted for publication once it meets all outstanding technical requirements.

Kind regards,

Wajid Mumtaz

Academic Editor

PLOS ONE

Additional Editor Comments (optional):

Reviewers' comments:

Reviewer's Responses to Questions

**Comments to the Author**

1. If the authors have adequately addressed your comments raised in a previous round of review and you feel that this manuscript is now acceptable for publication, you may indicate that here to bypass the “Comments to the Author” section, enter your conflict of interest statement in the “Confidential to Editor” section, and submit your "Accept" recommendation.

Reviewer #1: All comments have been addressed

2. Is the manuscript technically sound, and do the data support the conclusions?

Reviewer #1: Yes

3. Has the statistical analysis been performed appropriately and rigorously? 

Reviewer #1: Yes

4. Have the authors made all data underlying the findings in their manuscript fully available?

Reviewer #1: Yes

5. Is the manuscript presented in an intelligible fashion and written in standard English?

Reviewer #1: Yes

6. Review Comments to the Author

Reviewer #1: Thank you for this excellent contribution to the literature.

7. PLOS authors have the option to publish the peer review history of their article (what does this mean?). If published, this will include your full peer review and any attached files.

Reviewer #1: No

---

## [Editor Report · Acceptance letter]

27 Dec 2021

PONE-D-21-29049R1 

Unstructured clinical notes within the 24 hours since
admission predict short, mid & long-term mortality in adult
ICU patients 

Dear Dr. Mahbub:

I'm pleased to inform you that your manuscript has been deemed suitable for publication in PLOS ONE. Congratulations! Your manuscript is now with our production department. 

Kind regards, 

on behalf of

Dr. Wajid Mumtaz 

Academic Editor

PLOS ONE